# Molecular Classes and Growth Hormone Treatment Effects on Behavior and Emotion in Patients with Prader–Willi Syndrome

**DOI:** 10.3390/jcm11092572

**Published:** 2022-05-04

**Authors:** Ranim Mahmoud, Heidi D. Swanson, Merlin G. Butler, Pamela Flodman, June-Anne Gold, Jennifer L. Miller, Elizabeth Roof, Kathryn Osann, Elisabeth Dykens, Daniel J. Driscoll, Virginia Kimonis

**Affiliations:** 1Department of Pediatrics, University of California, Irvine, CA 92697, USA; dr_rony_2011@yahoo.com (R.M.); heidi_swanfell@hotmail.com (H.D.S.); pflodman@hs.uci.edu (P.F.); goldj@hs.uci.edu (J.-A.G.); 2Department of Pediatrics, Faculty of Medicine, Mansoura University, Mansoura 35516, Egypt; 3Childrens Hospital of Orange County, Orange, CA 92868, USA; 4Departments of Psychiatry & Behavioral Sciences and Pediatrics, University of Kansas Medical Center, Kansas City, KS 66160, USA; mbutler4@kumc.edu; 5Department of Pediatrics, Loma Linda University Medical School, Loma Linda, CA 92350, USA; 6Department of Pediatrics, College of Medicine, University of Florida, Gainesville, FL 32610, USA; millejl@peds.ufl.edu (J.L.M.); driscdj@peds.ufl.edu (D.J.D.); 7Vanderbilt Kennedy Center for Research on Human Development, Vanderbilt University, Nashville, TN 37235, USA; elizabeth.roof@vanderbilt.edu (E.R.); elisabeth.m.dykens@vanderbilt.edu (E.D.); 8Department of Statistics, University of California, Irvine, CA 92697, USA; kosann@uci.edu

**Keywords:** Prader–Willi syndrome, behavior, genetic subtypes, growth hormone

## Abstract

Prader–Willi syndrome (PWS) is a complex genetic disorder with three genetic classes. Patients with PWS are characterized by severe hypotonia, developmental delay, behavioral problems, learning disabilities and morbid obesity in early childhood if untreated. Data were collected through Rare Disease Clinical Research Network (RDCRN) from four study centers which evaluated patients with PWS. The Behavior Assessment System for Children 2nd edition (BASC-2) was chosen to provide behavioral assessment. Data from 330 participants ((64% 15q11-q13 deletion (DEL), 36% maternal disomy 15 (UPD)) were separated into three age groups and analyzed, 68% of whom were still actively receiving recombinant human growth hormone (rhGH) treatment. When comparing the BASC results by molecular subtype, parent-reported aggression was higher for the deletion than for the UPD cohort (*p* = 0.007). Participants who were on rhGH treatment showed lower scores for parent-reported hyperactivity and aggression (*p* = 0.04, 0.04, respectively), and a trend for anger control (*p* = 0.06) and teacher-reported attention problems and aggression (*p* = 0.01, 0.004, respectively). Additional adjusted analyses were undertaken and significant differences were noted in the GH versus non-GH treated groups for only teacher-reported aggression, which increased in the No GH treated patient group (*p =* 0.03). This study showed documented differences in PWS behavior by molecular class and rhGH treatment. RhGH therapy may be beneficial for certain behaviors in patients with PWS; however, observed differences need more studies for confirmation in the future.

## 1. Introduction

Prader–Willi syndrome (PWS) is a complex genetic disorder affecting 1/15,000–1/30,000 live births. PWS is characterized by hypotonia and poor appetite, which leads to feeding difficulty and poor weight gain in the newborn or infant stage. Then, it progresses after infancy to hyperphagia or excessive food drive, which can lead to obesity in childhood and beyond, if not externally controlled, (e.g., [1,2,3]).

Prader–Willi syndrome is a genetically heterogeneous disorder caused by the absence of paternal gene expression in the 15q11.2-q13 region. There are three main molecular classes. A paternal 15q11-q13 deletion (DEL) is the most common genetic cause seen in 60–70% of individuals. The less common form of PWS, in about 25–35% of cases, is caused by maternal uniparental disomy 15 (UPD) in which both copies of chromosome 15 have been inherited from the mother. A rare form occurring in the remaining 3–5% of PWS cases includes a defect of the imprinting center which controls the imprinted genes on chromosome 15 (e.g., [1,2,4]).

Patients with PWS have unique symptoms related to behavior, emotions, and learning, which are all part of a person’s phenotype [5]. Patients with PWS are often delayed in language and motor development, have learning difficulties and are easily frustrated, impulsive, quick to anger, stubborn and inflexible. They have a high pain threshold, are highly anxious and prone to skin-picking or other obsessive-compulsive behavior [5,6,7,8]. Attention deficit/hyperactivity symptoms and insistence on sameness are common with patients with PWS and have an early age of onset [5,6,7]. Butler et al. [8] also reported that patients with the PWS deletion were more affected than patients with UPD. Particularly, patients with the larger typical PWS type I deletion had more behavioral problems than those with the smaller typical type II deletion. Roof et al. [9] reported that patients with PWS UPD had significantly higher Verbal IQ scores than those with the deletion. Patients with the deletion had more self-injury and more severe behavioral problems than those with UPD [10]. Studies suggest that a characteristic behavioral pattern begins in early childhood in 70–90% of PWS patients. These patterns include temper tantrums, stubbornness, controlling and manipulative behavior, compulsive-like behaviors, such as skin-picking, and difficulty with changes in their routine [1,2,3,5,6,7,8,9,10,11,12]. The severity of behavioral problems increases with age [11] and then diminishes in older adults [11]. To better characterize emotion and behavior patterns in PWS, we examined a large cohort of PWS patients studied longitudinally from infancy through adulthood in a National Institutes of Health (NIH) funded PWS rare disease consortium involving four national centers specializing in PWS.

## 2. Materials and Methods

Data were collected through the use of the Rare Disease Clinical Research Network (RDCRN) Natural History of PWS study conducted at the University of Florida Health Science Center in Gainesville, Florida (lead site); University of California, Irvine; University of Kansas Medical Center, Kansas City, Kansas; and Vanderbilt University Medical Center in Nashville, Tennessee. This study was approved by the human subjects committee at each participating institution (e.g., University of California Irvine (UCI) Institutional Review Board (IRB) protocol number 2007-5605), and written informed consent was obtained from a parent or legal guardian. Data from 330 individuals with genetically confirmed PWS by molecular cytogenetic testing was utilized for this study [12,13]. Only subjects with a deletion or UPD were included in the analyses in this study due to the low number of patients with PWS in the imprinting defect class. Clinical, cognitive, behavioral, and PWS molecular genetic class data were collected over multiple visits in the 8-year longitudinal observational natural history study funded by the NIH.

The Behavior Assessment System for Children 2nd edition (BASC-2) was chosen to provide effective behavior assessment for ages 2 through 25 years which analyzes a person’s behavior from three perspectives: parent, teacher and self. The BASC-2 system has Parent Reporting Scales (PRS), Teacher Reporting Scales (TRS), Self-Report of Personality (SRP).

The items on the TRS and PRS can be used to calculate five composite scales: Adaptive Skills, a Behavioral Symptoms Index, Externalizing Problems, Internalizing Problems, and School Problems. The BASC-2 can be scored either by hand in about 30 min or by using the software profile. These scores are converted into T scores and percentiles [14]. It provided T-scores and percentiles for measurements of maladaptive behavior. High scores above 55 in the clinical scales suggest a person’s behavior is at risk, the one exception being leadership, a positive trait, where a low score indicates at-risk behavior. A T-score is a standardized score between 0 and 100. A score of 50 represents the mean and is considered average. A difference of 10 from the mean indicates a difference of one standard deviation.

The data were summarized using mean and standard deviation (SD) for continuous variables. Subject groups were subdivided by PWS molecular genetic classes and growth hormone use, duration, and onset, and then compared using two-group *t*-tests for continuous variables and chi-square tests for categorical variables. We categorized patients based on their treatment onset and duration with growth hormone as previously reported [12]. For this study, we only studied the cohort who were still currently taking recombinant human growth hormone (rhGH). RhGH use differed dramatically with the age of the patient. Because patients who used GH were significantly younger than those who did not use rhGH, it was necessary to explore the effect of age as a confounding variable. The effect of age was investigated by stratifying by age (≤11, 11–18, >18 years) and testing with multivariate analysis of variance (MANOVA). The statistical analyses were accomplished using Statistical Package for Social Sciences (SPSS) 20 Statistics software (Armonk, NY, USA). Statistical significance was considered at *p* < 0.01 because of multiple comparisons.

## 3. Results

### 3.1. Study Participants

The total number of 330 patients with PWS in this study included 148 males (45%) and 182 females (55%). There were 211 (64%) participants with 15q11.2-q13 deletions and 119 (36%) with UPD. Sixty-six percent were actively on growth hormone treatment (*n* = 219) with 42 percent of males (*n* = 92), and 58 percent of females (*n* = 127) mean dose of 0.22 mg/kg/wk. Eighty-five percent were ever on growth hormone treatment (*n* = 282) with 46 percent of males (*n* = 130), and 54 percent of females (*n* = 152). The average IQ score was 67 ± 16 with no difference identified by age groups with the majority of PWS participants found below 11 years (68.5%), 11 to 18 years (25.5%), and >18 years (6%) [13]. The average IQ in our study participants was very similar to what is reported in the literature in other PWS patient cohorts ( e.g., [9]).

The mean age for all study participants was 13.4 ± 1.5 years. The mean age at which growth hormone treatment began was 4.6 ± 7.2 years. The mean age at which growth hormone treatment was discontinued was 13 ± 8.9 years. Participants were evaluated for their behavior by parents, teachers and if applicable, self. Their mean scores are displayed in Table 1 with BASC scores for each behavior. The mean scores above 55 suggest at-risk behavioral problems, according to the BASC clinical scale. These include parent-reported attention problems, hyperactivity, aggression, and a trend for anger control. For the teacher-reported behaviors, hyperactivity, and leadership with a trend for anger control. However, the teacher-reported total leadership measure generated a score of 42.5 based on the BASC adaptive scale. A total score of 39.3 for a parent report would indicate that a lack of leadership may not be a behavioral problem in our participants. Age was separated into three categories (≤11 years, 11–18 years, >18 years). Significant differences were found between the three age groups for parent-reported attention problems (*p* = 0.030), parent-reported hyperactivity (*p* < 0.001) and parent-reported aggression (*p* = 0.012) while teacher-reported hyperactivity (*p* < 0.001) and teacher-reported leadership (*p* = 0.042) scores were also found. Better scores were seen in the younger age groups for both the parent and teacher assessments.

### 3.2. Comparison by Molecular Genetic Class (DEL vs. UPD)

The study participants were compared by their molecular genetic class (DEL or UPD). In this cohort, those with deletions were significantly older in age than those with maternal disomy (*t*-test; *p* = 0.032), with a mean age of 14.4 ± 12.1 years for those with DEL and 11.6 ± 10.1 years for those with UPD.

When comparing the BASC (behavior assessment) results by molecular genetic class, there was one significant difference in parent reporting of aggression with higher aggression in those with DEL versus for UPD (*p* = 0.007). However, DEL and UPD did not differ significantly with respect to the BASC categories: parent-reported attention problems, hyperactivity, leadership, or anger control; teacher-reported hyperactivity, aggression, leadership, or anger control, and self-reported attention problems (see Table 2). After adjusting for age using multivariate ANOVA methods, significant differences were found between UPD and DEL for parent-reported aggression (*p* = 0.007) with higher scores indicating more aggression in the DEL group. Worsening at-risk scores (i.e., 55 or greater) for attention problems, hyperactivity and anger control were observed in the parent assessments while attention problems, hyperactivity, and anger control were also observed in the teacher assessments.

### 3.3. Comparison of Growth Hormone Treatment

Mean ages for those on rhGH treatment were 9.3 years (SD = 7.7) with a range of two months to 49 years. Those not on rhGH were 21.8 years with a range of 2 to 62 years (SD = 13.4) (*p* < 0.0005). The mean BMI percentile was 84 ± 35 in the GH treated group vs. 85 ± 42 in the non-GH treated group (*p* = 0.51). Significant differences were also found between participants who received rhGH versus those who did not (with higher scores for those not on rhGH) for four of the twelve total BASC behavior variables (hyperactivity and aggression in parent assessments and attention and aggression in teacher assessments) and one trending variable (anger control in parent assessments) (Table 3). The mean scores for behavior for those not on rhGH showed consistently worsening scores for eight out of ten measures for the parent- and teacher-reported assessments with four being significant and one trending as noted.

### 3.4. Effect of Growth Hormone (GH) Treatment after Adjusting for Age in Patients with PWS

Because age differed significantly between those who used rhGH and those who did not use rhGH, additional analyses were done to adjust for this age difference when looking at differences in other variables. Age was separated into three categories (≤11 years, 11–18 years, >18 years). There was a significant difference in teacher-reported aggression (*p* = 0.033) between rhGH treated and untreated groups. Differences in parent-reported hyperactivity, parent-reported aggression, and teacher-reported attention problems by rhGH use were no longer statistically significant after adjusting for age (see Table 4).

## 4. Discussion

The purpose of this study was to describe differences in PWS behaviors by molecular class, and the use of GH treatment. This study was based on the largest existing dataset consisting of 330 patients with PWS enrolled from four national sites and similarly analyzed. We found a difference in the BASC scores between the molecular genetic classes, in the deletion group, parent-reported aggression was found to be significantly higher. This could be a real difference between the two molecular genetic classes (DEL or UPD), or it may be due to confounding variables like differences in age or the use of rhGH. Significant differences in the BASC scores were also found when we categorize the participants into three age groups with higher or worsening scores for parent-reported attention problems, hyperactivity and lower scores for aggression with advancing age in adulthood, while teacher-reported hyperactivity is higher in the 11–18 years age group compared to the group less than 11 years of age.

This study also identified significant differences in behaviors between those using and not using GH. They included parent-reported hyperactivity and aggression in addition to teacher-reported attention problems and aggression. However, after adjusting for age and grouping into three age groups, there were only significant differences with one of the BASC behavior results: teacher-reported aggression with higher scores with advanced age. It was found that out of the 24 different paired comparisons for Table 4 with adjustments for age only aggression rated by teacher showed the highest and most consistent abnormal scores for the three age groups in the No GH group.

GH has an effect on the promotion of growth and differentiation of the central nervous system and is an important component of proper cerebellar development and adult function. Children with GH deficiency show a decreased volume of brain structures such as the corpus callosum, hippocampus, thalamus, and basal ganglia, which correlate with cognitive and motor function deficits [15] and may lower aggression scores but more research is needed.

Use of GH appears to be associated with differences in behavior, but we can only speculate that GH use might be responsible for this difference since participant behavior was not assessed prior to and after GH use. Interpretation of results is difficult because of the age difference between the groups based on GH treatment. Patients treated with GH were younger than those not receiving GH, and even though adjustments were made for age in the statistical analysis, it is possible that age differences could contribute to the behavioral changes observed. This study supports previous controlled studies that showed growth hormone treatment is associated with behavior improvement [16,17,18,19]. One prior study concluded that if growth hormone treatment was started before the age of two years, there was a noticeable improvement in several variables [20]. The current findings suggest a potential benefit even if GH was not started until after the age of two years. Previous studies performed in adults with PWS, e.g., [21], suggest that specific behaviors like teacher-reported attention and teacher-reported aggression were normal in those who used GH in comparison to those who were not on GH treatment. In contrast, no improvement or deterioration of behavioral problems was seen in children with PWS during long-term GH treatment in an 8-year longitudinal study [22]. A meta-analysis by Luo et al. [23] reported that no behavioral improvement was identified after GH therapy. However, this assessment was based on questionnaires completed by parents; high parental expectations for behavioral improvement with GH treatment may have impacted the parental reporting in this study.

Lo et al. [24] reported in PWS that long-term treatment with GH has no improvement in behavioral problems but also no deterioration. Furthermore, Bohm et al. [25] reported that GH treatment has no improvement in behavioral problems but ceasing GH treatment led to marked behavioral deterioration. Donze et al. [26] reported improvement in both motor and mental development with GH treatment. Other studies reported that long-term GH treatment led to improvement of intelligence and cognitive development in patients with PWS [10,27].

In addition to GH treatment, there are other studies that indicate the efficacy of other treatment approaches on the behavioral phenotype. A pilot trial of diazoxide choline controlled-release tablets (DCCR) on behavior in patients with PWS reported dramatic improvements in aggressive, threatening and destructive behaviors with a decrease in self-injurious behavior [28]. In addition, a clinical trial to evaluate the effect of oxytocin treatment on behavior in children with PWS found lower scores for compulsive behaviors, decreased ritualistic behavior/insistence on sameness, decreased stereotypic behavior, and decreased restricted interests in the oxytocin treated group versus placebo requiring further testing for confirmation [29].

## 5. Conclusions

In summary, our study is one of the largest to date designed to assess the effects of molecular class and GH treatment on behavioral and emotional problems in patients with PWS having a wide age range. We found differences between DEL and UPD groups and in those treated with GH compared to those who did not receive GH treatment. The results suggest that the use of GH treatment should be considered soon after the diagnosis of PWS not only to increase stature and body composition but may positively affect behavior; however, further studies are needed with matching age at entry into the study, length of GH treatment with behavioral evaluation before and after GH treatment and adjustments for cognition at the onset of treatment, if needed, to confirm our observations.

## Figures and Tables

**Table 1 jcm-11-02572-t001:** Physical characteristics and behavior assessment scores using BASC in 330 genetically confirmed patients with PWS.

	Total		Age Groups						
	*n*	Mean ± SD	*n*	Age < 11 years Mean ± SD	*n*	Age 11–18 years Mean ± SD	*n*	Age > 18 years Mean ± SD	*p*-Value
Age (years)	330	13.4 ± 11.5	226	6 ± 2.5	84	14.3 ± 0.3	20	20.8 ± 0.5	<0.001
Age started rhGH (years)	219	4.6 ± 7.2	187	1 ± 2	22	4 ± 7.5	10	20.1 ± 0.5	0.001
BMI percentile for age and gender	330	82 ± 23	226	80 ± 21	84	85 ± 22		92.5 ± 2.5	0.421
BASC Results Statistics									
Parent-reported behaviors:									
Attention problems	212	56.9 ± 9.3	159	54.6 ± 2.6	33	56.4 ± 2.3	20	67.33 ± 4.3	0.030
Hyperactivity	213	56.8 ± 12.6	160	53.3 ± 3.3	33	64.6 ± 2.9	20	72.3 ± 2.3	<0.001
Aggression	213	50.8 ± 9.5	159	52.5 ± 3	34	55.3 ± 3	20	42.6 ± 2.6	0.012
Leadership	133	39.3 ± 10	92	45.3 ± 1.9	26	42.1 ± 2.4	15	40.3 ± 7.5	0.229
Anger control	100	56.7 ± 10.8	82	55.8 ± 3.4	10	64.8 ± 2.4	8	64 ± 2.6	0.064
Teacher-reported behaviors:									
Attention problems	153	56.8 ± 9.9	108	55.7 ± 3.4	30	52.8 ± 3.5	15	52.6 ± 7.3	0.477
Hyperactivity	155	56.8 ± 11	56	54 ± 2.5	84	62.2 ± 3.3	15	56.3 ± 8	<0.001
Aggression	155	55.5 ± 10.4	110	53.1 ± 2.5	36	58.7 ± 3.3	9	52.3 ± 4.6	0.156
Leadership	95	42.5 ± 7.7	63	46.7 ± 2.5	20	41.4 ± 2.3	12	48.3 ± 10.5	0.042
Anger control	71	57.4 ± 10.2	52	54.1 ± 2.3	12	62.9 ± 2.5	7	55 ± 10.5	0.064
Self-reported:									
Attention problems	46	51.5 ± 11.2	26	55.7 ± 3.4	12	52.8 ± 3.8	8	52.6 ± 7.3	0.157
Hyperactivity	63	50.0 ± 11.8	26	54.2 ± 3.8	29	48.2 ± 3	8	52 ± 4.3	0.377

BASC = The Behavior Assessment System for Children 2nd edition, PWS = Prader–Willi Syndrome, and SD = standard deviation. Mean scores above 55 suggest at risk behavioral problems.

**Table 2 jcm-11-02572-t002:** Demographics and behavior assessments by molecular class.

	Deletion			UPD			*p*-Value
	*n*	Mean	SD	*n*	Mean	SD	
Age (years)	209	14.4	12.1	118	11.6	10.1	
BMI percentile for age and gender	209	85	34	116	83	36	0.24
Parent-reported:							
Attention problems	134	56.8	9.7	78	57.1	8.7	0.82
Hyperactivity	135	57.2	13.3	78	55.9	11.1	0.48
Aggression	135	52.2	9.8	78	48.5	8.7	0.007
Leadership	87	39.4	10.4	46	39.2	9.4	0.93
Anger control	63	57.7	10.4	37	54.8	11.4	0.20
Teacher-reported:							
Attention problems	92	55.7	10.7	61	58.4	8.3	0.09
Hyperactivity	92	57.8	11.7	63	55.3	9.6	0.17
Aggression	92	56.1	10.8	63	54.5	9.9	0.34
Leadership	58	43.3	8.4	37	41.4	6.3	0.26
Anger control	43	57.0	11.5	28	57.9	8.1	0.72
Self-reported:							
Attention problems	32	52.2	12.2	14	49.9	8.6	0.53
Hyperactivity	43	51.7	12.3	20	46.2	9.8	0.08

UPD = maternal uniparental disomy 15, and SD = standard deviation. Mean scores above 55 suggest at risk behavioral problems.

**Table 3 jcm-11-02572-t003:** Demographics and behavior by growth hormone treatment in patients with PWS.

	GH			No GH			*p*-Value
	*n*	Mean	SD	*n*	Mean	SD	
Age (years)	219	9.3	7.7	101	21.8	13.4	<0.00
BMI percentile for age and gender	276	84	35	57	85	42	0.51
Parent-reported behaviors:							
Attention problems	168	57.0	9.6	40	56.5	8.1	0.78
Hyperactivity	168	55.8	12.4	41	60.3	13.1	0.04
Aggression	168	50.1	9.0	41	53.5	11.5	0.04
Leadership	94	38.7	9.4	37	41.2	11.5	0.21
Anger control	87	56.0	10.5	10	62.9	13.1	0.06
Teacher-reported behaviors:							
Attention problems	130	55.8	9.9	21	61.9	8.8	0.01
Hyperactivity	132	56.2	11.1	21	60.6	10.4	0.09
Aggression	132	54.5	10.3	21	61.5	9.9	0.004
Leadership	75	42.5	7.9	19	43.4	6.8	0.66
Anger control	67	57.3	10.3	3	60.3	13.5	0.62
Self-reported:							
Attention problems	33	51.8	11.8	12	51.3	9.9	0.89
Hyperactivity	50	50.3	12.4	12	49.3	9.5	0.78

PWS = Prader–Willi Syndrome, GH = growth hormone, and SD = standard deviation. Mean scores above 55 suggest at risk behavioral problems.

**Table 4 jcm-11-02572-t004:** Demographics and behavior by growth hormone use: Adjusted for age in patients with PWS.

	GH			No GH			*p*-Value
Age Groups	<11 Years	11–18 Years	>18 Years	<11 Years	11–18 Years	>18 Years	
Parent-reported behaviors:							
Attention problems	57 ± 10	60 ± 8.6	56 ± 14	57.5 ± 9	57.6 ± 7.6	49 ± 8.7	0.150
Hyperactivity	53 ± 10	66.3 ± 14.5	68 ± 7.4	56 ± 10	61 ± 9	60.7 ± 9	0.231
Aggression	48.9 ± 9	54.3 ± 11	52.2 ± 12.5	55.6 ± 16	53 ± 11	50 ± 5	0.638
Anger control	56 ± 10.5	59 ± 7	64 ± 1.1	60 ± 14	62 ± 14	61.5 ± 9	0.697
Teacher-reported behaviors:							
Attention problems	56.5 ± 9.1	57.4 ± 10.5	52 ± 12	56.7 ± 12	60 ± 11.3	50 ± 11.4	0.942
Hyperactivity	54 ± 9	64 ± 13	70 ± 10	56.5 ± 11	58 ± 10	57 ± 10	0.133
Aggression	51.6 ± 10	50 ± 12	53 ± 12	58.4 ± 13.5	59 ± 10	61.5 ± 7	0.033
Anger control	57 ± 9	56 ± 12	57 ± 10	57 ± 11	60 ± 8	59 ± 7	0.715

GH = growth hormone. Mean scores above 55 suggest at risk behavioral problems.

## Data Availability

The data presented in this study are not publicly available but are available from the corresponding author on reasonable request.

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
