# Peer review of "Molecular Classes and Growth Hormone Treatment Effects on Behavior and Emotion in Patients with Prader–Willi Syndrome"

_jcm, 2022, doi:10.3390/jcm11092572_

Round 1

Reviewer 1 Report

Dear Authors,

Thank you for including part of my suggestions for improvement of the manuscript. The most relevant issue from the first review remains:

“However, there are some main issues that need reconsideration, especially regarding the possible influence of recombinant human growth hormone treatment (rhGH).

In my opinion this analysis is not informative and does not give reliable conclusions mainly because of the age difference, what Authors have discussed in the paper, but also because of probably different age at the start of therapy, different time of the treatment duration. Moreover, I would suggest to compare patients with different molecular subgroups in regard to rhGH therapy.”

There are also some minor points, that I have not received an explanation (as below, part of the first review):

  1. Line 46: “Prader-Willi syndrome is a genetically heterogeneous disorder caused by loss or absence of paternal gene expression in the 15q11.2-q13 region.”- I am not sure how to understand “loss or absence”.
  2. Line 84: Could the Authors refer in more detail to the method- why is it t-scores, not z-
  3. Lines 119-120: “However, teacher reported leadership measures with a score of 5 based on the BASC adaptive scale, and 39.3 for parent report would indicate that a lack of leadership may not be a behavioral problem.”- I am afraid this conclusion is unclear.
  4. Line 163: “Age was separated into two categories based on median age for this analysis: young = ≤11 and old = >11 y.”- the data in the manuscript are given as the mean values. If the Authors decide to use a median for splitting the study population, the median (min-max) values should be presented before.
  5. The division of the study group into 2 groups adjusting for age regarding the median age is difficult to I would recommend to adjust the group according to the age but in a smaller samples and separately for the children, adolescents and adults. As discussed at the beginning of the review, otherwise we cannot give any reliable conclusions.

Again, I appreciate the Authors effort but I would still recommend to split the manuscript. The part regarding rhGH treatment does not provide relevant conclusions at the moment.

Author Response

Dr. Emmanuel Andrès                                                                                

Editor-In-Chief

Department of Internal Medicine,

University Hospital of Strasbourg,

67000 Strasbourg, France

Dear Dr. Andrès,

We thank you for permitting us to make revisions to further strengthen our manuscript for publication. We appreciate the detailed critiques of the two reviewers and have provided a point-by-point response to the concerns in the cover letter and in the revised manuscript.

Reviewer 1

Thank you for including part of my suggestions for improvement of the manuscript. The most relevant issue from the first review remains:

“However, there are some main issues that need reconsideration, especially regarding the possible influence of recombinant human growth hormone treatment (rhGH).

In my opinion this analysis is not informative and does not give reliable conclusions mainly because of the age difference, what Authors have discussed in the paper, but also because of probably different age at the start of therapy, different time of the treatment duration. Moreover, I would suggest to compare patients with different molecular subgroups in regard to rhGH therapy.”

There are also some minor points, that I have not received an explanation (as below, part of the first review):

  1. Line 46: “Prader-Willi syndrome is a genetically heterogeneous disorder caused by loss or absence of paternal gene expression in the 15q11.2-q13 region.”- I am not sure how to understand “loss or absence”.

Response: The text was changed to Prader-Willi syndrome is a genetically heterogeneous disorder caused by absence of paternal gene expression in the 15q11.2-q13 region.’

  1. Line 84: Could the Authors refer in more detail to the method- why is it t-scores, not z-scores

Response: We used BASC score second edition which is a series of instruments for children, adolescents, and young adults between the ages of 2 and 25. The BASC-2 system has Parent Rating Scales (PRS), Teacher Rating Scales (TRS), Self-Report of Personality (SRP).

The items on the TRS and PRS can be used to calculate five composite scales: Adaptive Skills, a Behavioral Symptoms Index, Externalizing Problems, Internalizing Problems, and School Problems. SRP can also be used to calculate composite scales as The Emotional Symptoms Index, Inattention/Hyperactivity, Internalizing Problems, and Personal Adjustment,

The BASC-2 can be scored either by hand in approximately 30 min or by using the ASSIST Plus software profile. The raw scores are transformed into T scores and percentiles (Reference 12).

  1. Lines 119-120: “However, teacher reported leadership measures with a score of 5 based on the BASC adaptive scale, and 39.3 for parent report would indicate that a lack of leadership may not be a behavioral problem.”- I am afraid this conclusion is unclear.

Response: The mean scores above 55 suggest at-risk behavioral problems, according to the BASC clinical scale. In our participant the teacher reported leadership measures generated a score of 42.5 based on the BASC adaptive scale.  The parent reported leadership score was 39.3so this is not a behavioral problem in our patients with PWS as these scores are less than 55.

  1. Line 163: “Age was separated into two categories based on median age for this analysis: young = ≤11 and old = >11 y.”- the data in the manuscript are given as the mean values. If the Authors decide to use a median for splitting the study population, the median (min-max) values should be presented before.

Response: The study was divided into three age groups children, adolescents and adults (≤11 years, 11-18 years, >18 years). Data are now presented as mean and standard deviation

  1. The division of the study group into 2 groups adjusting for age regarding the median age is difficult to I would recommend to adjust the group according to the age but in a smaller samples and separately for the children, adolescents and adults. As discussed at the beginning of the review, otherwise we cannot give any reliable conclusions.

Response: The study was divided into three age groups (≤11 years, 11-18 years, >18 years). In agreement with the reviewer ‘s suggestion the study was divided into three age groups (≤11 years, 11-18 years, >18 years). We have developed and generated data into the text and revised tables to account for three age groups as suggested and summarized our findings accordingly within the revised tables and text of the manuscript throughout in order to strengthen the manuscript

Again, I appreciate the Authors effort but I would still recommend to split the manuscript. The part regarding rhGH treatment does not provide relevant conclusions at the moment.

   Response:  Teacher reported aggression showed highest and abnormal scores for the three age groups in the No GH group. So, GH treatment may lower aggression scores, but more research is needed. Also, our study solidifies observations from prior studies that rhGH is not associated with adverse findings.

Yours Sincerely, 

Virginia Kimonis, MD

Univ. of California-Irvine Med. Center

Mail: 101 The City Drive South, ZC4482,

Orange CA 92868

Tel: +1 (714) 456-5791; Fax: +1 (714)456-5330, Pager +1 (714) 506-2063

Email: vkimonis@uci.edu

Reviewer 2 Report

I think the manuscript has been improved and I'm pleased with the responses and corrections.

I think there is a mistake in Results, in lines 123-124:

"Eighty-five percent were ever on growth hormone 123 treatment (N= 282) with 88 percent of males (N=130), and 83 percent of females (N=152)"

Author Response

Dr. Emmanuel Andrès                                                                                

Editor-In-Chief

Department of Internal Medicine,

University Hospital of Strasbourg,

67000 Strasbourg, France

Dear Dr. Andrès,

We thank you for permitting us to make revisions to further strengthen our manuscript for publication. We appreciate the detailed critiques of the two reviewers and have provided a point-by-point response to the concerns in the cover letter and in the revised manuscript.

Reviewer 2

comments and Suggestions for Authors

I think the manuscript has been improved and I'm pleased with the responses and corrections.

I think there is a mistake in Results, in lines 123-124:

"Eighty-five percent were ever on growth hormone 123 treatment (N= 282) with 88 percent of males (N=130), and 83 percent of females (N=152)"

Response: The percentages were corrected in the manuscript males (N=130) 46 percent, and 54 percent of females (N=152)

Yours Sincerely, 

Virginia Kimonis, MD

Univ. of California-Irvine Med. Center

Mail: 101 The City Drive South, ZC4482,

Orange CA 92868

Tel: +1 (714) 456-5791; Fax: +1 (714)456-5330, Pager +1 (714) 506-2063

Email: vkimonis@uci.edu

Round 2

Reviewer 1 Report

In my opinion the Authors improved the manuscript according to the main recommandations.

This manuscript is a resubmission of an earlier submission. The following is a list of the peer review reports and author responses from that submission.

Round 1

Reviewer 1 Report

I think that You have undertaken an important subject. I am impressed with the big study group. 

These are my comments on the manuscript:

  1. The average dose of recombinant growth hormone during therapy could be given.
  2. There is a lack in the discussion to try to explain how genetics or growth hormone can influence the behavior of the PWS patients - the possible mechanism. The influence of GH on the central nervous system and psychiatric status may be interesting for the reader.
  3. The number of treated patients with GH is 219 in Results, but Tables 3 and 4 differ (218; 2017). I don't understand. 
  4. There are some patients who are not actively treated with GH but were treated in the past. It could influence the behavior as well. I think it should be discussed. 
  5. Is it possible that the stage of mental retardation could influence the children's behavior? 

Reviewer 2 Report

Thank you for the opportunity to review the manuscript “Molecular Classes and Growth Hormone Treatment Effects on Behavior & Emotion in Prader-Willi Syndrome” by Ranim Mahmoud et al. It is an interesting and detailed analysis of differences in behavioural problems in patients with PWS with different molecular subtypes.

However, there are some main issues that need reconsideration, especially regarding the possible influence of recombinant human growth hormone treatment (rhGH).

In my opinion this analysis is not informative and does not give reliable conclusions mainly because of the age difference, what Authors have discussed in the paper, but also because of probably different age at the start of therapy, different time of the treatment duration. Moreover, I would suggest to compare patients with different molecular subgroups in regard to rhGH therapy.

Please find my other comments below.

  1. Title: Molecular Classes and Growth Hormone Treatment Effects on Behavior & Emotion in Prader-Willi Syndrome- I would suggest modifying the title: “patients with Prader-Willi syndrome”
  2. I would recommend the use of a term: recombinant human growth hormone (rhGH).
  3. Abstract: “Prader-Willi syndrome (PWS) is a complex genetic disorder with three genetic classes. PWS patients are characterized by severe hypotonia, developmental delay, behavioral problems, learning disabilities and morbid obesity in early childhood.”- again please use a term “patients with PWS” in the whole manuscript; it is necessary to add: “if untreated”.
  4. Line 46: “Prader-Willi syndrome is a genetically heterogeneous disorder caused by loss or absence of paternal gene expression in the 15q11.2-q13 region.”- I am not sure how to understand “loss or absence”.
  5. Line 47: “There are three difference molecular classes.”- rather “three main”, the possible translocation mechanism should be mentioned. Moreover, the percentage should be described in ranges.
  6. Line 57: “They have a high pain threshold, highly anxious and prone to skin-picking or other obsessive-compulsive behavior.”- unclear; please refer to the studies regarding differences between molecular subtypes.
  7. Line 75: “UCI IRB protocol number 2007-5605”- please explain the abbreviation.
  8. Line 84: Could the Authors refer in more detail to the method- why is it t-scores, not z-scores.
  9. Lines 119-120: “However, teacher reported leadership measures with a score of 42.5 based on the BASC adaptive scale, and 39.3 for parent report would indicate that a lack of leadership may not be a behavioral problem.”- I am afraid this conclusion is unclear.
  10. Results- the data are unnecessary repeated in the text and in the tables.
  11. Line 163: “Age was separated into two categories based on median age for this analysis: young = ≤11 y. and old = >11 y.”- the data in the manuscript are given as the mean values. If the Authors decide to use a median for splitting the study population, the median (min-max) values should be presented before.
  12. The division of the study group into 2 groups adjusting for age regarding the median age is difficult to understand. I would recommend to adjust the group according to the age but in a smaller samples and separately for the children, adolescents and adults. As discussed at the beginning of the review, otherwise we cannot give any reliable conclusions.
  13. The list of references is too short.

In conclusion, I appreciate the Authors effort. However, I would recommend maybe split the manuscript into two. First part, regarding the molecular subtypes, seems to be more reliable but there is a need for widening the introduction, discussion, references.

The second part, regarding rhGH treatment, definitely needs more thought and improvement.